# Psychosocial Determinants of COVID-19 Vaccine Hesitancy and the Mediating Role of Various Attitudes towards Science

**DOI:** 10.3390/vaccines11081310

**Published:** 2023-07-31

**Authors:** Jonathan Morgan, Joseph A. Wagoner, Tom Pyszczynski

**Affiliations:** Psychology Department, University of Colorado, 1420 Austin Bluffs Pkwy, Colorado Springs, CO 80919, USAtpyszczy@uccs.edu (T.P.)

**Keywords:** COVID-19 vaccine hesitancy, attitudes towards science, social psychology, political ideology, science and religion, reactance

## Abstract

This study examined the way attitudes towards science in the U.S. mediate the relationship between COVID-19 vaccine hesitancy and psychosocial predictors, such as political ideology, religiosity, reactance proneness, dogmatism, perceived communal ostracism, education, and socioeconomic status. We analyzed the structure of people’s attitudes towards science, revealing four distinct factors: epistemic confidence, belief that science and technology are beneficial, trust in science in general, and trust in medical science. With all four factors included as mediators in a saturated path analysis, low levels of trust in medical science and low epistemic confidence fully mediated the relationships between nearly all of the psychosocial predictors and COVID-19 vaccine hesitancy. Political conservativism’s negative association with vaccine hesitancy was partially mediated by the same two facets of people’s attitudes towards science. Adding nuance to existing research, we found that trust in science in general was not a significant mediator once all four facets were included in the model. These findings are discussed with a focus on their implications for understanding attitudes towards science and their substantial and complex role in COVID-19 vaccine hesitancy.

## 1. Introduction

Even as the pandemic wanes, understanding why such a large percentage of Americans responded to the vaccines with hesitancy remains important for shaping our response to future health crises. Prior to COVID-19, substantial research across disciplines showed that vaccine hesitancy, the refusal or delayed acceptance of vaccination despite availability, is multifaceted and will vary by time, place, and the type of vaccine [1]. For example, risk perceptions [2], sense of collective responsibility [3], and structural constraints [4] will all shape an individual’s hesitancy [5].

COVID-19 vaccine hesitancy in particular appears to depend on a unique constellation of psychosocial antecedents. For example, people’s experiences with long-COVID [6], their fear of vaccination [7], and their beliefs about the collective importance of vaccination [8] all impact their level of acceptance of the COVID-19 vaccine.

As social psychologists have examined this unique constellation of psychosocial determinants in the United States of America (USA), they found that Americans’ attitudes towards science [9] and their political leaning [10] are consistently strong predictors of COVID-19 vaccine hesitancy. Other researchers have highlighted the specific role that religiosity [11], reactance proneness [12], and structural constraints [13] play in COVID-19 vaccine hesitancy. For the most part, however, each of these determinants is considered by itself.

This study aims to integrate these various streams of research by investigating the role that attitudes towards science play in mediating the relationship between vaccine hesitancy and political ideology along with these other prominent predictors. We also differentiate various aspects of people’s attitudes towards science so that future efforts to successfully advocate for vaccine uptake can target the relevant aspects of these attitudes. Given the complexity of both vaccine hesitancy and attitudes towards science, we suggest that this focused approach to attitude change is likely to be more successful than general calls for trusting science.

### 1.1. Determinants of Attitudes towards Science and COVID-19 Vaccine Hesitancy

In order to better understand the mediating role played by attitudes towards science, we focus on psychosocial factors that are likely to predict both COVID-19 vaccine hesitancy and these attitudes.

#### 1.1.1. Political Ideology

Political conservativism in the USA has been a significant predictor of negative general attitudes towards science for decades [14], though this association varies depending on the specific scientific topic [15]. Prior to COVID-19, the specific association between conservativism and vaccine hesitancy was mixed, with some studies finding liberals more hesitant [16] and others suggesting that conservatives were less likely to be vaccinated [17]. The link between political conservativism and COVID-19 vaccine hesitancy, however, is well established [18,19]. Underscoring the significance of this relationship, Sehgal and colleagues found that deaths rates from COVID-19 were higher in predominantly conservative than liberal counties across the USA, and vaccine uptake explained 10% of this difference [20]. Nevertheless, it is important to note that a majority of Republicans are not hesitant about COVID-19 vaccination [18], so understanding the factors that explain why some conservatives accept and others refuse the COVID-19 vaccines remains critically important.

#### 1.1.2. Religiosity

Similarly, high levels of religiosity in the USA are associated with negative attitudes towards science [21], but this association likely depends on the type of religiosity and the specific scientific issue [22]. This complexity is reflected in work on the relationship between religiosity and COVID-19 vaccine hesitancy. Some studies show a negative association between religiosity and intention to vaccinate [11,23]. But other work that takes political ideology into account found that religious service attendance was positively associated with vaccination, and that it was Christian nationalism that accounted for the negative association with COVID-19 vaccination rates [24]. It may not be religiosity per se but particular forms of politicized and/or reactive religiosity that drives COVID-19 vaccine skepticism [25]. Similar to political ideology, it is important to note that in each of these studies, the majority of religious participants were either vaccinated or expressed the intention to be vaccinated, which suggests that other psychosocial factors are involved.

#### 1.1.3. Reactance Proneness

Reactance, the motivational state to restore a sense of freedom after experiences of restrictions on one’s behavior [26], is one such variable. Health psychologists have tracked how individual differences in reactance proneness predict resistance to public health messages [27]. Following this trend, amidst the pandemic, various studies have found a link between reactance proneness and fewer preventative health behaviors, including vaccination [12,28]. An open question is whether this link is influenced by the complex relationship between reactance and attitudes towards science [29].

#### 1.1.4. Dogmatism

Another potentially influential factor in COVID-19 vaccine hesitancy is dogmatism—the “unchangeable and unjustified certainty in one’s beliefs” [30]. Research suggests that dogmatic tendencies to reject new information and opposing evidence predict more negative attitudes towards science [31,32]. Throughout the pandemic, especially regarding the novel vaccines, COVID-19 science has required accommodating new information and potentially shifting one’s beliefs, yet few, if any, studies have examined the role of dogmatism in COVID-19 vaccine hesitancy.

#### 1.1.5. Perceived Communal Ostracism

Broadening our scope, we also suggest that experiences of ostracism may influence both negative attitudes towards science and COVID-19 vaccine hesitancy. Gauchat found that experiences of exclusion are an important driver of science skepticism [33]. This is underscored by Harambam and Aupers’ study, which found that skeptics of science tend to “feel excluded, mocked, and stigmatized” [34]. This potential negative feedback cycle is reflective of experiences of ostracism [35], which can lead to a wide range of negative behaviors, including health behaviors [36]. Political polarization in the USA raises the possibility that experiences of one’s group as being ostracized may drive both skepticism towards science and the COVID-19 vaccines in much the same way that personal ostracism leads to less cooperation and more risky behaviors [37].

#### 1.1.6. Structural Constraints

Finally, there are also likely to be some individuals who are unvaccinated due to structural constraints, such as low socioeconomic status (SES) [4] or low levels of education [38]. For example, consistent predictors of low intention to vaccinate throughout the pandemic have been lack of access [13] and low SES [39]. An open question, however, is whether this relationship is influenced by broader attitudes towards science.

#### 1.1.7. Covariance among Predictors

Many of the psychosocial factors outlined above are related. For example, political conservatism tends to be associated with religiosity [40], reactance [41], and dogmatism [42]. Similarly, high levels of religiosity in the USA have been associated with low SES [43], dogmatism [44], and, potentially, reactance [45]. Given the high likelihood of significant relationships among these predictors, any public health efforts to encourage vaccination would benefit from understanding how each relates to COVID-19 vaccine hesitancy when this shared variance is controlled.

### 1.2. Attitudes towards Science

Understanding the potential mediating role that attitudes towards science may play in influencing COVID-19 vaccine hesitancy requires clarifying what we mean by these attitudes. Many studies operationalize these attitudes by focusing on specific topics, such as childhood vaccination [46] or climate change [16] for example. This approach clarifies specific stances around contentious topics but cannot provide a strong inference about the way these particular attitudes may influence other specific issues, such as COVID-19 vaccine hesitancy.

Other studies assess broad attitudes towards science using items from national surveys, such as the General Social Survey [14], the National Science Foundation’s Science Indicators Survey [33], or the British Social Attitudes Survey [47]. This approach helpfully sheds light on general positions but is limited in its capacity to illuminate the structure of these broad attitudes and connect them to specific stances.

A third approach tends to focus on participants’ trust towards science. For example, Sturgis and colleagues found that trust in science, at a societal level, predicted vaccine confidence across the globe [9]. But this approach typically deploys single item measures of trust, which preclude a deeper understanding of the way trust can be multidimensional and vary across disciplines [48].

This taxonomy of methods for studying attitudes towards science is not comprehensive but instead focuses on the most prominent ways to assess these attitudes. Within our study, we draw together these methods in order to assess whether they index a common factor or multiple facets of people’s attitudes towards science. If the latter, then it is crucial to understand which facets of these attitudes best predict vaccine hesitancy and how they may be mediating the relationship between the psychosocial determinants outlined above and vaccine hesitancy.

### 1.3. The Current Study

The current study examines the extent to which these psychosocial variables predict people’s COVID-19 vaccine hesitancy through their attitudes towards science. In order to account for likely covariance among these psychosocial variables, we use a fully saturated path analysis, with psychosocial variables as predictors, attitudes towards science as mediators, and COVID-19 vaccine hesitancy as the outcome variable.

While portions of this model are exploratory by nature, our primary hypotheses follow the literature outlined above:Stronger political conservativism will be related to more COVID-19 vaccine hesitancy.Stronger religiosity will be related to more COVID-19 vaccine hesitancy.Higher levels of reactance proneness will be related to more vaccine hesitancy.Higher levels of dogmatism will be related to more COVID-19 vaccine hesitancy.Lower levels of Perceived SES, will be related to more vaccine hesitancy.Finally, more perceived communal ostracism will be related to more COVID-19 vaccine hesitancy.Attitudes towards science will mediate the relationships between predictor variables and COVID-19 vaccine hesitancy.

The specific facets of participants’ attitudes towards science that will mediate these relationships remain exploratory since we have not yet conducted the necessary factor analysis. The impact that controlling for covariance among all predictors will have on these specific relationships likewise remains largely exploratory.

## 2. Materials and Methods

The study was approved by the UCCS IRB.

### 2.1. Sample

During December 2021, we recruited participants through Prolific Academic, an online platform that allows researchers to recruit and compensate people for completing studies. Prolific Academic users have more diverse characteristics and produce higher quality data than other crowdsourcing sites or college samples [49]. We limited our sample to people living in the U.S. and recruited an equal number of vaccinated and unvaccinated participants.

Using semPower in R [50], we conducted a power analysis for an SEM with 40 degrees of freedom, RMSEA of 0.05, α = 0.01, and power = 0.95, which suggested a sample of 397 participants. We doubled this amount and added ~10% for possible failed attention checks, resulting in a total of 900 participants (450 from each sub-population).

Our final sample size was 921, with 458 of them unvaccinated at the time of data collection. The average age was 36 years old (*SD* = 13.9), with 449 females, 454 males, and 18 people choosing neither male nor female. Our sample was predominately White Americans (*n* = 639; Black/African American *n* = 68; Hispanic/Mexican American *n* = 56; East Asian *n* = 31; South Asian *n* = 20; Middle Eastern *n* = 7; Native American *n* = 5; Multiple-ethnicity or no response *n* = 95).

### 2.2. Procedure and Materials

The surveys were presented in two separate blocks using Qualtrics. After consenting, participants completed questionnaires for reactance proneness, dogmatism, and communal ostracism in a counterbalanced order, each on its own page. After this first block, participants completed questionnaires assessing COVID-19 attitudes, attitudes towards science, and trust in science, which were also counterbalanced to prevent any order effects. After completing both blocks, participants answered a few sociodemographic questions, including SES, political orientation, and religiosity. Finally, they were debriefed before being redirected back to Prolific.co for payment. All materials for the study are available in the Appendix A, along with the exploratory analyses mentioned below.

### 2.3. Predictors

#### 2.3.1. Reactance Proneness

We used Hong and Faedda’s 11-item reactance scale (revised), which gauges both the cognitive and affective dimensions of reactance [51]. Items were assessed on a 5-point Likert scale (1 = strongly disagree, 5 = strongly agree), with higher scores indicating more proneness to reactance (α = 0.84).

#### 2.3.2. Dogmatism

We used Altemeyer’s 20-item DOG scale to assess dogmatic beliefs [52]. Each item is scored on a 9-point Likert scale from −4 (strongly disagree) to +4 (strongly agree), with 0 as a neutral position. Higher scores indicate more of a tendency to hold beliefs dogmatically (α = 0.93).

#### 2.3.3. Perceived Communal Ostracism

We created a novel set of eight questions focusing on facets of perceived communal ostracism that may impact an individual’s attitudes towards science: being ignored, excluded, and mocked—for example, “my community is excluded from many of the important decisions being made today”. Each item was measured on a 7-point Likert scale (1 = strongly disagree, 7 = strongly agree). See factor analysis below.

#### 2.3.4. Attitudes toward Science

To assess attitudes towards science, we used a combination of items from previous studies. Sturgis and Allum used four items to assess attitudes towards science and technology [47]. We used all four of these items but adjusted “We depend too much on scientific expertise and not enough on faith”, replacing “faith” with “personal experience” to avoid the forced dichotomy between religion and science. We also included 2 items from Gauchat’s operationalization of attitudes towards science: “Science and technology will create more opportunities for the next generation” and “On balance, the benefits of scientific research have outweighed the harmful results” [33].

In order to assess different aspects of these attitudes towards science, we also included a set of four questions that reflect the epistemic concerns of skeptics surfaced by Harambam and Aupers [34]. These questions focused on perceived dogmatism, exclusion, and both personal and financial bias within science.

All items were scored on a 5-point Likert scale with higher scores reflecting more positive attitudes towards science. See factor analysis below.

#### 2.3.5. Trust in Science

We used a two-part questionnaire to measure trust in science. The first part assessed trust in science in general as well as across disciplines: “science in general; physicists; chemists, biologists; psychologists; social scientists”. Participants rated each group on an 11-point Likert scale (0 = Do not trust at all, 10 = Trust completely). The second part of the questionnaire focused on medical scientists, starting again at the general level and then moving through various facets of medical science: “medical science in general; your family doctor; public health researchers; epidemiologists; public health institutions (e.g., NIH or the CDC); and pharmaceutical researchers”. These items were also measured on an 11-point Likert scale. See factor analysis below.

#### 2.3.6. COVID-19 Vaccine Attitudes and Status

We assessed vaccine status through a single item asking participants “as of today, have you been vaccinated for COVID-19?” Based on participants’ responses, we asked how hesitant do [did] you feel about getting vaccinated?” (1 = not at all, 7 = very hesitant).

#### 2.3.7. Political Ideology

As a measure of political ideology, we used a single item within our sociodemographic survey, which asked participants where they place themselves on the political spectrum (1 = very liberal, 7 = very conservative).

#### 2.3.8. Religiosity

We measured religiosity using a single item asking participants about the importance of their religiosity to them (1 = not at all important, 5 = extremely important). This question was part of our sociodemographic survey.

#### 2.3.9. Structural Constraints: Education and SES

Our sociodemographic survey also included questions about gender, age, highest level of educational attainment (1 = grade school; 2 = some high school; 3 = high school grad; 4 = some college; 5 = college grad; 6 = Master’s degree; 7 = PhD) and perceived SES (1 = worst off, 10 = best off).

## 3. Results

The first step in our analysis examined the structure of participants’ attitudes toward and trust in science and the structure of perceived communal ostracism through exploratory factor analyses. Using these factors, we analyzed the role they play in mediating the relationship between our psychosocial predictors and COVID-19 vaccine hesitancy by conducting a saturated path analysis using the lavaan package (version 0.6–120 [53] in R (version 4.2.1).

### 3.1. Factor Analyses

#### 3.1.1. Attitudes towards and Trust in Science

We ran an exploratory factor analysis that included all 10 items from our attitudes towards science questionnaire along with the 12 trust in science items. Given the high likelihood that our factors will correlate, we used oblimin rotation. This analysis suggested a five-factor structure accounting for 67% of the total variance (Factor 1: 18%, eigenvalue = 3.87; Factor 2: 16%, eigenvalue = 3.55; Factor 3: 15%, eigenvalue = 3.22; Factor 4: 10%, eigenvalue = 2.15; Factor 5: 9%, eigenvalue = 2.01). For more detailed statistics, see Appendix A.

Factor 1 included four items: trust in science in general, along with trust towards physicists, chemists, and biologists; factor loadings ranged from 0.63 to 0.89 (α = 0.96). Factor 2 included all six items about trust in medical science, with factor loadings ranging from 0.47 to 0.75 (α = 0.93). Factor 3 included our four novel items assessing the belief that science is biased or objective, along with the question about depending too much on expertise. Factor loadings ranged from 0.58 to 0.74 (α = 0.89). Factor 4 included the four items drawn from Gauchat [33] and Sturgis and Allum [47] that focus on the belief that science and technology are beneficial. Factor loadings ranged from 0.43 to 0.78 (α = 0.75). Finally, factor 5 included the two items about levels of trust towards psychologists and social scientists, with loadings of 0.82 and 0.83, respectively. The implication that trust towards social scientists and psychologists is distinct from trust in science in general is a result worth pursuing (see Appendix A for analyses along these lines). In our subsequent analyses, however, we focus solely on the first four factors since these have stronger conceptual links with vaccine hesitancy. One item, “It is not important for me to know about science in my daily life” did not load well on any of the factors (all loadings < 0.24) so was excluded from analyses.

#### 3.1.2. Perceived Communal Ostracism

Since our questionnaire for communal ostracism was novel, we conducted an exploratory factor analysis with varimax rotation. Two factors accounted for 60% of the total variance (Factor 1: 17%, eigenvalue = 1.38; and Factor 2: 43%, eigenvalue = 3.42). Item loadings were all >0.50. Factor 1 consisted of only two items, both of which express the experience of being left behind as the world changes: e.g., “I feel my community and those I hold dear are being left behind as society changes”. Factor 2 includes the other six items that focus on experiences of perceived exclusion, denigration, and being ignored. Since the experience of being left behind is conceptually distinct from that of communal ostracism, which is our primary focus, we excluded those two items and used the mean of the remaining six as our variable for perceived communal ostracism (α = 0.89).

### 3.2. Main Analyses

#### Bivariate Correlations

Table 1 reports the bivariate Pearson’s correlations between all variables without controlling for covariance. All our predictors and all facets of participants’ attitudes towards science are significantly associated with COVID-19 vaccine hesitancy.

Given the high number of significant correlations among our psychosocial predictors and mediators, we conducted a saturated path analysis to account for covariance in these relationships. This enabled us to assess the unique association between each predictor and vaccine hesitancy when variance shared with all other predictors is controlled and to assess the hypothesized mediating role of the various aspects of attitudes toward science in the relationship of the other predictors with vaccine hesitancy.

### 3.3. Path Analysis

Figure 1 presents the significant associations from our path analysis, with all path coefficients standardized. Since our model was fully saturated, there are no fit indices to report. All variables are included as observed variables, as operationalized above. Table 2 includes all of the coefficients and test statistics from the path analysis. This full model explained 51% of variance in COVID-19 vaccine hesitancy.

All path coefficients are standardized. For the sake of clarity, only those associations that are statistically significant are reported in the diagram. The model ran with all potential associations. For a full list of covariances, see Table 2.

#### 3.3.1. Predicting COVID-19 Vaccine Hesitancy from Psychosocial Variables

The total effect (c-paths) of each predictor on vaccine hesitancy, without including any mediators, is reported in Table 2. While education, dogmatism, and perceived communal ostracism all had significant bivariate correlations with COVID-19 vaccine hesitancy, these associations became non-significant when accounting for shared variance among predictors. Of the predictors, political conservativism was the most strongly associated with higher levels of COVID-19 vaccine hesitancy. Even after accounting for political orientation, COVID-19 vaccine hesitancy was still significantly predicted by higher reactance proneness, lower SES, and stronger religiosity.

#### 3.3.2. Predicting Attitudes towards Science

The strongest independent predictors of all facets of participants’ attitudes towards science are political orientation and reactance proneness. Higher levels of religiosity are moderately associated with weaker belief in scientific objectivity (*p* < 0.001) and less trust in science in general (*p* < 0.001), but the negative association with the belief that science and technology are beneficial is only marginal (*p* = 0.02). Education on the other hand is marginally associated with beliefs that science and technology are beneficial (*p* = 0.044) and more trust in science in general (*p* < 0.001).

We calculated the percentage of variance explained by subtracting the remaining variance for each facet of attitudes towards science from 1. Our psychosocial predictors explained the following: 40% of the variance in the belief that science is objective (*p* < 0.001); 29% of the variance in the strength of belief that science and technology are beneficial (*p* < 0.001); 27% of the variance in trust towards science in general (*p* < 0.001); and 34% of the variance in trust towards medical science (*p* < 0.001).

#### 3.3.3. Predicting Vaccine Hesitancy from Attitudes towards Science

Not all facets of participants’ attitudes towards science equally predicted COVID-19 vaccine hesitancy. High levels of trust in medical science were, unsurprisingly, the most strongly and negatively associated with vaccine hesitancy (*p* < 0.001). The belief that science is objective held its own unique negative association with vaccine hesitancy (*p* < 0.001). With these two facets of attitudes towards science accounted for, the strength of belief that science and technology are beneficial was only marginally associated with vaccine hesitancy (*p* = 0.013), and trust in science in general was not significantly related to vaccine hesitancy (*p* = 0.09).

#### 3.3.4. Mediation

With the mediators included in our path analysis (see Table 3), the direct effects of reactance proneness and religiosity on vaccine hesitancy were non-significant (*p* = 0.100 and *p* = 0.080, respectively), suggesting full mediation by various attitudes towards science.

SES and political orientation, on the other hand, retained significant direct effects on COVID-19 vaccine hesitancy (*p*s < 0.001). The more strongly participants identified as conservative, the more hesitant they were about the COVID-19 vaccines, even while accounting for their various attitudes towards science. While this relationship remained significant, the standardized coefficient dropped from *β* = 0.46 to *β* = 0.25 (see Table 2), suggesting partial mediation. The small negative association between SES and COVID-19 vaccine hesitancy remained relatively constant regardless of the mediators.

Table 3 reports a full test of the indirect effects within our path analysis. Given the b-paths reported above, it is unsurprising that trust in medical science was the primary mediator for many of the relationships between our psycho-social predictors and COVID-19 vaccine hesitancy. The positive relationship with political orientation, where higher levels of conservativism predicted greater COVID-19 vaccine hesitancy, is partially mediated through low levels of trust in medical science (*p* < 0.001) and low belief that science is objective (*p <* 0.001) but not by levels of trust in general science (*p* = 0.099) and only marginally by the degree of belief that science and technology are beneficial (*p* = 0.018).

Similarly, the positive relationships between COVID-19 vaccine hesitancy and reactance proneness, dogmatism, and perceived communal ostracism were fully mediated by low levels of trust in medical science (*p*s ≤ 0.001) and weaker belief that science is objective. The mediating effect of how much participants believed that science and technology are beneficial was only marginal, and the level of trust in science more generally had no mediating effect on COVID-19 vaccine hesitancy.

In contrast, religiosity’s positive association with COVID-19 vaccine hesitancy appears to be solely mediated by low levels of the belief that science is objective (*p* = 0.001), not any of the other facets of attitudes towards science (*p*s ≥ 0.089). The negative association between COVID-19 vaccine hesitancy and SES, on the other hand, was solely mediated by trust in medical science (*p* = 0.007), not the other aspects of participants’ attitudes towards science (*p*s ≥ 0.290).

## 4. Discussion

The major goals of this study were to shed light on the relationships between psychosocial predictors of COVID-19 vaccine hesitancy, with a focus on the mediating role of several distinct aspects of attitudes toward science. High levels of COVID-19 vaccine hesitancy were significantly and uniquely associated with more political conservativism (H1), more reactance proneness (H3), lower SES (H5), and more religiosity (H2). These associations were either fully or partially mediated by people’s attitudes towards science (H7), which factor analyses differentiated into four factors: trust in medical science; trust in science in general; the belief that science is objective; and the belief that science and technology are beneficial. This multidimensional perspective on attitudes towards science is helpful because our psychosocial variables had distinct relationships with each of the different facets. More importantly, when all four facets of these attitudes were included in our analyses, only trust in medical science and the belief that science is objective were reliably negatively associated with vaccine hesitancy. General trust in science showed no relationship with vaccine hesitancy once these other facets were accounted for.

Our findings nuance previous research suggesting that trust in science is a crucial determinant for COVID-19 vaccine beliefs and behaviors [9]. Echoing Larson et al.’s point, the way researchers have measured trust in science varies quite widely [48]. Our results suggest that if distinctions are made between different facets of people’s attitudes towards science, then it is not trust in science in general that predicts COVID-19 vaccine hesitancy. Trust in medical science is conceptually closer to our outcome variable, vaccine hesitancy, than trust in general science, which could partially explain why it was the stronger predictor. But the value of including both in the model is apparent when considering their relationship with religiosity, which was negatively associated with general trust but not related to trust in medical science. This distinction can help public health efforts that are focused on building trust in science in general by specifying the aspects of science that different populations may find objectionable [55].

From this multidimensional perspective on attitudes towards science, the strongest predictors of COVID-19 vaccine hesitancy were low trust in medical science and weak belief that science is objective. This suggests that people have a complex understanding of how science operates and distinguish between the aspects of science most relevant to the topic at hand. Reiterating this point, the belief that science and technology are beneficial was only marginally related to COVID-19 vaccine hesitancy.

Understanding the content and structure of individuals’ attitudes towards science provides more actionable and specific points of engagement than working to build trust in science in general. For example, speaking to specific epistemic concerns about bias and objectivity in pharmaceutical science may be more effective for softening vaccine hesitancy than abstract discussions about the role of science in society. Additional research is needed to provide solid grounding for policy recommendations.

Keeping these different facets of attitudes towards science in mind can also help unravel some of the complex relationships between psychosocial factors and vaccine hesitancy. For example, our results suggest that a portion of the relationship between religiosity and COVID-19 vaccine hesitancy is likely accounted for by political conservativism, which considerably overlaps with religiosity in the USA [56], and the rest of this relationship is explained by the low levels of belief that science is objective among some religious individuals. Importantly, the connection between religiosity and COVID-19 vaccine hesitancy does not appear to be due to low levels of trust in science in general or medical science in particular. Again, this nuanced perspective can help address vaccine hesitancy among certain groups by directing communication efforts towards the particular concern and away from reifying narratives about growing antiscientific attitudes.

While trust in medical science and belief in scientific objectivity fully mediated most of the relationships between psychosocial factors and vaccine hesitancy, high levels of political conservativism remained moderately associated with more hesitancy. Though our mediation analyses suggest that nearly half of this relationship is accounted for by lower levels of trust in medical science and less belief that science is objective, that leaves the other half unexplained.

The significant positive relationships between political conservativism and religiosity, reactance proneness, and dogmatism may also account for some its connection with vaccine hesitancy. But within our path analysis, when this shared variance was controlled, it did not substantially change the relationship between political conservativism and vaccine hesitancy. This may reflect the ways in which COVID-19 vaccination has been politicized beyond science into a social and economic issue that is part of people’s political identity [18].

Undoubtedly, political orientation is important for understanding COVID-19 vaccine hesitancy, but it is not the full story. In our bivariate correlations, every psychosocial variable was significantly related to vaccine hesitancy. This initial finding corroborates work on the role of reactance [28], religiosity [11], education, and SES (Al-Jayyousi et al., 2021) within COVID-19 vaccine hesitancy while also widening this line of research to include dogmatism (H4) and perceived communal ostracism. Once shared variance among these psychosocial variables was accounted for in the path analysis, reactance proneness, religiosity, and SES retained significant associations with COVID-19 vaccine hesitancy. These findings suggest that among individual difference factors, political orientation may be the strongest predictor, but it is not alone in driving vaccine hesitancy.

For example, individuals across the political spectrum who are prone to reactance are likely to be more hesitant about the vaccine. Low levels of trust in medical science and low belief that science is objective fully explained this relationship. Prior research on this relationship with COVID-19 vaccine hesitancy has examined how reactance proneness interacts with perceptions of threat [12], message fatigue [57], and a variety of other individual differences such as moral concerns and disgust sensitivity [28]. Our study corroborates and extends this work by suggesting that for individuals who are reactance prone, their hesitancy towards the COVID-19 vaccine arises in large part due to their perception of medical science as not a trustworthy source of information and their sense of bias within scientific inquiry. Given the central role of trust in persuasion, this result suggests a potential negative feedback loop may occur when sharing prescriptive medical information with people who are sensitive to any perceived impingement on their freedom.

We predicted that experiencing communal ostracism would lead to vaccine hesitancy (H6) in much the same way that experiences of individual ostracism led to antisocial and self-defeating behaviors [37]. The association between perceived communal ostracism and vaccine hesitancy became nonsignificant once we accounted for shared variance among predictors. This suggests that any connection between perceived communal ostracism and vaccine hesitancy is likely explained by SES and/or reactance proneness, both of which were moderately associated with perceived communal ostracism in our path analysis.

Finally, our results show that the relationship between SES and vaccine hesitancy was not substantially impacted by any of the other psychosocial variables. This echoes findings from early in the pandemic and across the globe [39] and suggests that COVID-19 vaccine hesitancy is at least partially shaped by structural constraints, such as economic insecurity. In our study, SES was positively associated with trust in medical science, but this relationship did not mediate its negative association with vaccine hesitancy. Research has shown that physicians in the U.S. tend to perceive low-income patients in a negative light [58], which in turn impacts their clinical decisions [59]. Low income patients, in turn, recognize the way SES impacts the care they receive [60], which may explain the positive association between SES and trust in medical science. This lack of trust, however, did not account for the negative relationship between SES and vaccine hesitancy, suggesting that financial concerns may have driven perceptions of the COVID-19 vaccine [61].

### Limitations

The primary limitation of this study is that it is cross-sectional in design. This precludes any causal inferences about the relationships found. The theoretical framework guiding our path analysis assumes that individual differences shape attitudes, which then shape action [62]. Nevertheless, our results are not causal. A second potential limitation is that our sample was recruited online. While studies have found that online samples provide reliable quality data [49], this recruitment method may lead us to underestimate the role that structural constraints, such as lacking internet access, may play in vaccine hesitancy. Finally, our study was specifically focused on these relationships within the USA. Given the context sensitivity of vaccine hesitancy and the unique political environment of the USA, our results cannot be reliably extended to other cultural environments.

## 5. Conclusions

This study examined some of the psychosocial determinants of COVID-19 vaccine hesitancy along with the mediating role that people’s attitudes towards science play in these dynamics. Political conservativism, reactance proneness, religiosity, and lower SES were all significantly related to more vaccine hesitancy. Participants’ attitudes towards science fully mediated the relationships between vaccine hesitancy and both reactance proneness and religiosity. These attitudes only partially mediated the relationships between vaccine hesitancy and political conservativism and SES.

Importantly, our analyses suggest that attitudes towards science are not monolithic entities. Instead, they can be differentiated into trust in medical science; trust in science in general; the belief that science is objective; and the belief that science and technology are beneficial. Trust in medical science and beliefs about scientific objectivity were the facets of these attitudes most predictive of COVID-19 vaccine hesitancy. If we hope to mitigate vaccine hesitancy by working to shift these attitudes, then we will likely be better off with a nuanced approach that focuses on specific facets of these attitudes rather than attempting to increase trust in science at a general level.

## Figures and Tables

**Figure 1 vaccines-11-01310-f001:**
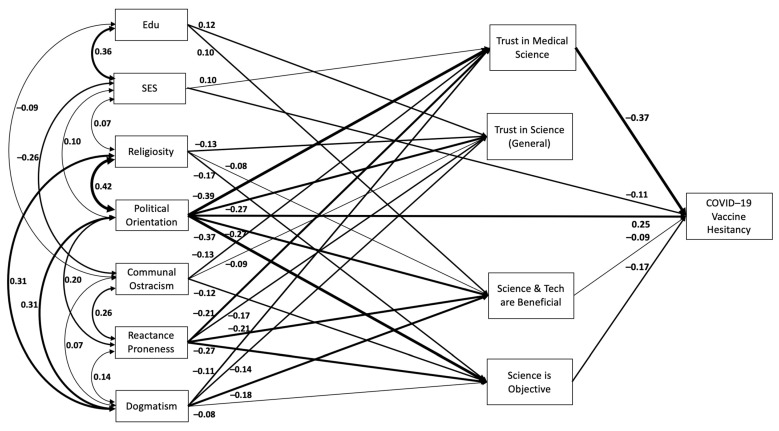
Full Model.

**Table 1 vaccines-11-01310-t001:** Bivariate Pearson Correlations Between all Variables.

Variable	1	2	3	4	5	6	7	8	9	10	11	12
1. Edu	4.4 (1.0)											
2. SES	0.35 ***	5.0 (1.7)										
3. Rel	−0.02	0.06	2.3 (1.5)									
4. RP	−0.05	−0.03	0.00	2.9 (0.7)								
5. Dog	−0.03	0.02	0.31 ***	0.16 ***	4.1 (1.3)							
6. ComOst	−0.09 **	−0.25 ***	0.02	0.23 ***	0.08 *	4.0 (1.3)						
7. Pol	−0.02	0.09 *	0.38 ***	0.20 ***	0.31 ***	−0.02	3.4 (1.8)					
8. SciObj	0.08 *	0.02	−0.34 ***	−0.39 ***	−0.30 ***	−0.19 ***	−0.51 ***	3.3 (1.0)				
9. STBen	0.14 ***	0.07 *	−0.25 ***	−0.32 ***	−0.34 ***	−0.12 ***	−0.40 ***	0.68 ***	4.0 (0.8)			
10. TrSci	0.17 ***	0.08 *	−0.29 ***	−0.27 ***	−0.31 ***	−0.12 ***	−0.39 ***	0.62 ***	0.60 ***	7.2 (2.3)		
11. TrMed	0.14 ***	0.12 ***	−0.24 ***	−0.35 ***	−0.29 ***	−0.19 ***	−0.46 ***	0.68 ***	0.61 ***	0.82 ***	6.4 (2.4)	
12. VaxHes	−0.15 ***	−0.14 ***	0.29 ***	0.29 ***	0.25 ***	0.11 ***	0.53 ***	−0.58 ***	−0.51 ***	−0.52 ***	−0.64 ***	3.9 (2.5)

* *p* < 0.05; ** *p* < 0.01; *** *p* < 0.001; Mean (Standard deviations) reported along the diagonal. Ranges: Edu = Education (1–7); SES = Perceived Socioeconomic Status (1–10); Rel = Religiosity (1–5); RP = Reactance Proneness (1–5); Dog = Dogmatism (1–9); ComOst = Perceived Communal Ostracism (1–7); Pol = Political Orientation (1–7); SciObj = Belief that Science is Objective (1–5); STBen = Belief that Science and Technology are beneficial (1–5); TrSci = Trust in General Science (0–10); TrMed = Trust in Medical Science (0–10); VaxHes = COVID-19 Vaccine hesitancy (1–7).

**Table 2 vaccines-11-01310-t002:** Path Analysis between Predictors, ATS Mediators, and COVID-19 vaccine hesitancy.

Variables	B	SE	z	Semi-r_p_^2^	β	*p*
Outcome: COVID-19 Vaccine Hesitancy (Without Mediators)						
Education	−0.11	0.07	−1.50	<0.01	−0.05	0.134
SES	−0.20	0.05	−4.38	0.02	−0.14	<0.001
Religiosity	0.16	0.06	2.87	0.01	0.09	0.004
RP	0.62	0.11	5.76	0.03	0.18	<0.001
Dogmatism	0.09	0.06	1.40	<0.01	0.04	0.161
Com Ost	0.05	0.06	0.91	<0.01	0.03	0.366
Political Orientation	0.63	0.05	13.99	0.16	0.46	<0.001
Outcome: Belief that Science is Objective						
Education	0.04	0.03	1.27	<0.01	0.04	0.203
SES	<0.01	0.02	−0.02	<0.01	−0.00	0.986
Religiosity	−0.12	0.02	−5.33	0.03	−0.17	<0.001
RP	−0.41	0.05	−8.97	0.07	−0.27	<0.001
Dogmatism	−0.07	0.03	−2.75	0.01	−0.08	0.006
Com Ost	−0.09	0.02	−3.83	0.01	−0.12	<0.001
Political Orientation	−0.22	0.02	−11.41	0.10	−0.37	<0.001
Outcome: Belief that Science and Tech are Beneficial						
Education	0.07	0.02	2.89	0.01	0.10	0.004
SES	0.01	0.02	0.93	<0.01	0.03	0.352
Religiosity	−0.04	0.02	−2.32	0.01	−0.08	0.020
RP	−0.23	0.04	−6.39	0.04	−0.21	<0.001
Dogmatism	−0.11	0.02	−5.45	0.03	−0.18	<0.001
Com Ost	−0.03	0.02	−1.61	<0.01	−0.05	0.106
Political Orientation	−0.11	0.02	−7.57	0.05	−0.27	<0.001
Outcome: Trust in Science in General						
Education	0.25	0.07	3.55	0.01	0.12	<0.001
SES	0.06	0.05	1.35	<0.01	0.05	0.176
Religiosity	−0.21	0.05	−3.81	0.01	−0.13	<0.001
RP	−0.55	0.11	−5.17	0.03	−0.17	<0.001
Dogmatism	−0.25	0.06	−4.16	0.01	−0.14	<0.001
Com Ost	−0.15	0.06	−2.68	0.01	−0.09	0.007
Political Orientation	−0.33	0.04	−7.48	0.09	−0.27	<0.001
Outcome: Trust in Medical Science						
Education	0.12	0.07	1.63	<0.01	0.05	0.104
SES	0.13	0.05	2.92	0.01	0.10	0.003
Religiosity	−0.07	0.06	−1.31	<0.01	0.04	0.191
RP	−0.72	0.11	−6.73	0.04	−0.21	<0.001
Dogmatism	−0.21	0.06	−3.55	0.01	−0.11	<0.001
Com Ost	−0.22	0.06	−4.03	0.01	−0.13	<0.001
Political Orientation	−0.51	0.05	−11.47	0.11	−0.39	<0.001
Outcome: COVID-19 vaccine hesitancy						
Education	−0.05	0.06	−0.73	<0.01	−0.02	0.465
SES	−0.15	0.04	−3.76	0.01	−0.11	<0.001
Religiosity	0.09	0.05	1.74	<0.01	0.05	0.080
RP	0.16	0.10	1.65	<0.01	0.05	0.100
Dogmatism	−0.04	0.06	−0.65	<0.01	−0.02	0.515
Com Ost	−0.07	0.05	−1.34	<0.01	−0.04	0.181
Political Orientation	0.35	0.04	7.89	0.04	0.25	<0.001
Sci Objective	−0.41	0.10	−4.09	0.01	−0.17	<0.001
Sci & Tech Beneficial	−0.30	0.12	−2.49	<0.01	−0.09	0.013
Trust in General Sci	0.09	0.05	1.69	0.01	0.08	0.090
Trust in Medical Sci	−0.38	0.05	−7.47	0.04	−0.37	<0.001

sr^2^ = squared semi-partial correlations, representing the proportion of variance in each outcome that can be uniquely explained by each predictor while accounting for the other predictors [54].

**Table 3 vaccines-11-01310-t003:** Mediation Analyses.

Indirect Effects	β	SE	z	*p*
Education				
Edu → SciObj → Vax hes	−0.01	0.01	−1.22	0.224
Edu → S&T Bene → Vax hes	−0.01	0.01	−1.89	0.059
Edu → Tr Sci → Vax hes	0.01	0.01	1.54	0.127
Edu → Tr Med → Vax hes	−0.02	0.03	−1.59	0.112
SES				
SES → SciObj → Vax hes	0.00	0.01	−0.02	0.986
SES → S&T Bene → Vax hes	−0.00	0.00	−0.87	0.384
SES → Tr Sci → Vax hes	0.00	0.01	1.06	0.290
SES → Tr Med → Vax hes	−0.04	0.02	−2.72	0.007
Religiosity				
Rel → SciObj → Vax hes	0.03	0.02	3.25	0.001
Rel → S&T Bene → Vax hes	0.01	0.01	1.70	0.089
Rel → Tr Sci → Vax hes	−0.01	0.01	−1.55	0.122
Rel → Tr Med → Vax hes	0.02	0.02	1.29	0.197
Reactance Proneness				
RP → SciObj → Vax hes	0.05	0.04	3.72	<0.001
RP → S&T Bene → Vax hes	0.02	0.03	2.32	0.020
RP → Tr Sci → Vax hes	−0.01	0.03	−1.61	0.108
RP → Tr Med → Vax hes	0.08	0.06	5.00	<0.001
Dogmatism				
Dog → SciObj → Vax hes	0.01	0.01	2.28	0.023
Dog → S&T Bene → Vax hes	0.02	0.01	2.27	0.023
Dog → Tr Sci → Vax hes	−0.01	0.01	−1.57	0.117
Dog → Tr Med → Vax hes	0.04	0.03	3.21	0.001
Perceived Communal Ostracism				
Com Ost → SciObj → Vax hes	0.02	0.01	2.80	0.005
Com Ost → S&T Bene → Vax hes	0.01	0.01	1.36	0.175
Com Ost → Tr Sci → Vax hes	−0.01	0.01	−1.43	0.152
Com Ost → Tr Med → Vax hes	0.05	0.02	3.55	<0.001
Political Orientation (towards Conservativism)				
Pol → SciObj → Vax hes	0.06	0.02	3.85	<0.001
Pol → S&T Bene → Vax hes	0.03	0.01	2.37	0.018
Pol → Tr Sci → Vax hes	−0.02	0.02	−1.65	0.099
Pol → Tr Med → Vax hes	0.14	0.03	6.26	<0.001

## Data Availability

The data presented in this study are available on request from the corresponding author. The data are not publicly available due to privacy restrictions.

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
