# Peer review of "Psychosocial Determinants of COVID-19 Vaccine Hesitancy and the Mediating Role of Various Attitudes towards Science"

_vaccines, 2023, doi:10.3390/vaccines11081310_

Round 1

Reviewer 1 Report

This study explores psycho-social factors that are associated with COVID-19 vaccine hesitancy. Although the efficacy of the vaccine has been demonstrated, many people hesitated to have vaccine. This study is expected to provide interesting information. However, some problems hinder this study and authors must improve their work intensively.

“Introduction” is the section that describes the background and purpose of the study. These information are not available from this manuscript. Instead, explanations about factors that are expected to be associated with COVID-19 vaccine hesitancy are too long. Most of them are not necessary and should be deleted. Concise expression is necessary.

Table 2: Data on “Outcome: COVID19 Vaccine Hesitancy (without mediators)” is necessary?

It is difficult to understand results of path analysis. I guess the most important part is result of COVID-19 Vaccine Hesitancy with the meditators included in the path analysis. But it sounds that results of path analysis for mediators are more important. They can be greatly condensed.

Discussion p13 The sentence “past results using monolithic, single item, measures of attitudes towards science could be misleading” : Reference for this comment should be cited.

“Of course”: This idiom must be changed in expression suited for scientific article.

Conclusions: This section should summarize the overall findings but conclusion section of this manuscript does not. The conclusion section should not repeat background information from the “Introduction” but this manuscript does. The authors need to learn how to write the Conclusion section.

Author Response

Please see the attachment. Our responses are highlighted using track-changes. Thank you for your time and attention.

Reviewer 2 Report

Dear authors, hereafter some suggestions to improve the paper:

Abstract and keywords are not reported.

Section 1. Introduction

This section should be expanded to provide a more comprehensive background of the phenomenon under study. The introduction lacks a specific definition of vaccine hesitancy provided by MacDonald (2015), and important works to define it in the broad context of vaccination (e.g., Betsch et al., 2018) are missing.

I would suggest to describe a general perspective and contextualisation of vaccine hesitancy and psychological factors known to correlate before reporting the literature on the factors of interest in the following sections. For example, considering the impact of the long-Covid experience of vaccine hesitancy (Duradoni, Gursesli et al., 2022), the role of trust and attitudes towards medicine (Freeman et al., 2022) and vaccines (Santavicca et al., 2023), and the importance of vaccination fear (Duradoni, Veloso et al., 2022).

References:

Betsch, C., Schmid, P., Heinemeier, D., Korn, L., Holtmann, C., & Böhm, R. (2018). Beyond confidence: Development of a measure assessing the 5C psychological antecedents of vaccination. PloS one, 13(12), e0208601.

Duradoni, M., Gursesli, M. C., Materassi, L., Serritella, E., & Guazzini, A. (2022). The Long-COVID Experience Changed

People’s Vaccine Hesitancy but Not Their Vaccination Fear. International Journal of Environmental Research and Public Health, 19(21), 14550.

Duradoni, M., Veloso, M. V., La Gamma, M., Monciatti, A. M., & Guazzini, A. (2022). Italian version of the Vaccination

Fear Scale (VFS-6): internal and external validation. Mediterranean Journal of Clinical Psychology, 10(2).

Freeman, D., Loe, B. S., Chadwick, A., Vaccari, C., Waite, F., Rosebrock, L., ... & Lambe, S. (2022). COVID-19 vaccine

hesitancy in the UK: the Oxford coronavirus explanations, attitudes, and narratives survey (Oceans) II. Psychological medicine, 52(14), 3127-3141.

MacDonald, N. E. (2015). Vaccine hesitancy: Definition, scope and determinants. Vaccine, 33(34), 4161-4164.

Santavicca, T., Ngov, C., Frounfelker, R., Miconi, D., Levinsson, A., & Rousseau, C. (2023). COVID-19 vaccine hesitancy among young adults in Canada. Canadian Journal of Public Health, 114(1), 10-21.

Section 1.1

This section should briefly recall the ideological, psychological and social factors explored in the following parts of the paper to maintain greater cohesion with the rest of the text.

Section 1.1.1

I would suggest including in this section a brief introduction to the related 1.1.1.1 and 1.1.1.2 similarly to what is done for section 1.1.2 and 1.1.3.

Section 1.2.1.

On line 4, next to the acronym GMOs, the corresponding meaning should be added, as the topic but not the abbreviation has been mentioned before. Similarly for abbreviations on line 9 (e.g.,

GSS, NSF).

Section 1.2.2

This section should explain the research hypotheses in a clear and detailed manner, including the expected association between each predictor and vaccine hesitancy, and the hypothesised mediating role of the factors of interest. I would also suggest highlighting direct relations between the hypothesis and the literature presented.

Section 2.2

At line 7 the acronym SES should be explained.

Table 2

I would suggest replacing the .00 outcomes with <.01.

Sections 3.3.2., 3.3.3, 3.3.4.

The results are clearly presented. However, I would recommend adding the p-value at the end of each results explained.

Section 4

The discussion is supported by the results and highlights comparisons with previous literature.

However, there is a need for an explanation of the study limitations and potential biases.

I would also suggest resuming and deepening the discussion about future implications that the study might suggest to address vaccine hesitancy in light of the observed results.

Author Response

Please see the attachment. Our responses are highlighted using track-changes. Thank you for your time and attention, we believe that your review has significantly improved our manuscript, so thank you for the help.

Round 2

Reviewer 1 Report

Some problems that were pointed out in previous review are not revised. These comments are issued to improve the state of manuscript as much as possible. High quality studies are made by resolving problems. Authors are required to address modestly instead of insisting their opinions.

Introduction is a section that includes background and purpose of the study and not the wide knowledge of technical terms. Long explanation for determinants obscures the real purpose of the study. The authors should compile why political ideology, religiosity, reactance proneness, dogmatism and perceived communal ostracism concisely in one paragraph. The content written in 2 pages is not required by readers.

I have no idea what meaning the description about “precarity” has in this study. It is not included in determinants. This causes confusion.  

What is the meaning of the paragraph of Integrated procedure”?

The “Introduction” section is needed to be extensively revised to be easily understood.

There is no description about data on Political ideology is acquired in Materials & Methods.

In the previous review, I pointed out that description in “3.3.2. Predicting attitudes towards science” is too long. It looks like as if main finding of this study. You can describe only significant associations between variables. You can explain the results without mentioning all data. It is a common way that is seen in many articles. Balancing is important.

Author Response

Some problems that were pointed out in previous review are not revised. These comments are issued to improve the state of manuscript as much as possible. High quality studies are made by resolving problems. Authors are required to address modestly instead of insisting their opinions.

  • Thank you for your comments and feedback on our article. You’ve helpfully point out some of the shortcomings and tangents in our presentation, especially within the introduction. We have worked to significantly shorten and focus this material. Hopefully you, and others, find that it succinctly describes both the background and purpose of each variable included in the study. We’ve included more specific responses below.

Introduction is a section that includes background and purpose of the study and not the wide knowledge of technical terms. Long explanation for determinants obscures the real purpose of the study. The authors should compile why political ideology, religiosity, reactance proneness, dogmatism and perceived communal ostracism concisely in one paragraph. The content written in 2 pages is not required by readers.

  • We have significantly revised the introduction so as to focus narrowly on the relationship each determinant has with attitudes towards science and COVID-19 vaccine hesitancy. While editing, we recognized that we were giving a much broader context for each determinants’ relationship with attitudes towards science (e.g., associations with attitudes towards climate change or GMOs) and that this information could be obscuring rather than helpful. We’ve accordingly trimmed all of this content and reduced each determinant down to one paragraph as advised.
  • We preserved the introductory paragraph that gives the technical definition for vaccine hesitancy along with some of the key studies since this was added at the recommendation of the other reviewer and we belief helps to contextualize this study.
  • We hope that these revisions make the purpose of the study more easily understood, while also providing the necessary background to explain why each variable is included.

I have no idea what meaning the description about “precarity” has in this study. It is not included in determinants. This causes confusion. 

  • We have replaced the technical word “precarity” with either low SES or economic instability throughout the manuscript.

What is the meaning of the paragraph of Integrated procedure”?

  • We changed the title of this section to the more precise: “Covariance among predictors.” We also added references to studies that document the significant relationships among the determinants included in this study. Hopefully this helps make clear that part of our purpose is to control for these relationships in order to provide a more precise picture of the unique variance each has with vaccine hesitancy and attitudes towards science.

The “Introduction” section is needed to be extensively revised to be easily understood.

  • We hope that the revisions described above help make the purpose and background for this study more easily understood.

There is no description about data on Political ideology is acquired in Materials & Methods.

  • We previously had a single section for our sociodemographic survey, which included political orientation, religiosity, education, and SES. To more clearly orient the reader, we’ve made separate sections 2.3.7-2.3.9.

In the previous review, I pointed out that description in “3.3.2. Predicting attitudes towards science” is too long. It looks like as if main finding of this study. You can describe only significant associations between variables. You can explain the results without mentioning all data. It is a common way that is seen in many articles. Balancing is important.

  • We have trimmed this section down so that it does not repeat all of the associations reported in Table 2. We have also removed the descriptions of results that were not significant. Hopefully this strikes a helpful balance between the full description of the results in Table 2 and the verbal description in 3.3.2.

Reviewer 2 Report

Authors improved the quality of the paper that reached, in my opinion, the suitability to be published.

Author Response

Thank you again for your feedback. We believe your comments and suggestions significantly improved the article and are glad to find that you agree.

This round of revisions was done in response to suggestions from the other reviewer. While revising we've maintained your revisions from round 1.

Thank you again and best wishes